# A Computational Model for the Simulation of Prepulse Inhibition and Its Modulation by Cortical and Subcortical Units

**DOI:** 10.3390/brainsci14050502

**Published:** 2024-05-15

**Authors:** Thiago Ohno Bezerra, Antonio C. Roque, Cristiane Salum

**Affiliations:** 1Center of Mathematics, Computation and Cognition, Universidade Federal do ABC, São Bernardo do Campo 09606-045, Brazil; 2Department of Physics, School of Philosophy, Sciences and Letters of Ribeirão Preto, University of São Paulo, Ribeirão Preto 14040-901, Brazil; 3Interdisciplinary Applied Neuroscience Unit, Universidade Federal do ABC, São Bernardo do Campo 09606-045, Brazil

**Keywords:** computational neuroscience, prepulse inhibition, acoustic startle response

## Abstract

The sensorimotor gating is a nervous system function that modulates the acoustic startle response (ASR). Prepulse inhibition (PPI) phenomenon is an operational measure of sensorimotor gating, defined as the reduction of ASR when a high intensity sound (pulse) is preceded in milliseconds by a weaker stimulus (prepulse). Brainstem nuclei are associated with the mediation of ASR and PPI, whereas cortical and subcortical regions are associated with their modulation. However, it is still unclear how the modulatory units can influence PPI. In the present work, we developed a computational model of a neural circuit involved in the mediation (brainstem units) and modulation (cortical and subcortical units) of ASR and PPI. The activities of all units were modeled by the leaky-integrator formalism for neural population. The model reproduces basic features of PPI observed in experiments, such as the effects of changes in interstimulus interval, prepulse intensity, and habituation of ASR. The simulation of GABAergic and dopaminergic drugs impaired PPI by their effects over subcortical units activity. The results show that subcortical units constitute a central hub for PPI modulation. The presented computational model offers a valuable tool to investigate the neurobiology associated with disorder-related impairments in PPI.

## 1. Introduction

The acoustic startle response (ASR) is characterized by a motor reflex in response to a sudden high intensity and salient sound (called pulse) [1]. This reflex is modulated by the sensorimotor gating, a function of nervous system that prevents the brain from an overflow of information by avoiding interference of other stimuli during information processing [2]. The prepulse inhibition (PPI) phenomenon is an operational measure of sensorimotor gating, defined as the reduction in ASR amplitude to a pulse when this is preceded, in milliseconds, by a weaker stimulus (the prepulse). The prepulse on its own does not generate an ASR response.

Brainstem nuclei are the core regions involved in the mediation of ASR and PPI. Electrophysiological, pharmacological, and lesion experiments conducted in animals [1,3,4,5] showed that the cochlear root nucleus, the caudal pontine reticular nucleus, and the motor neurons in the spinal cord mediate the ASR. Similarly, the inferior and superior colliculi and the pedunculopontine tegmental nucleus are responsible for the inhibition of ASR circuit and generating the PPI [5,6]. More recent works with PET scan in rats corroborated that these same units are involved in the mediation of ASR and PPI [7,8].

Other studies showed that the cortical and subcortical regions are associated with the modulation of both ASR and PPI. Injection of a GABAergic agonist into the amygdala in rats [9] or optogenetic activation of the amygdala in mice [10] and muscimol injection into the rat nucleus accumbens [11] led to deficits in the PPI. PET scan studies conducted with rats also showed the involvement of the prefrontal cortex, hippocampus, and ventral pallidum [7,8]. Similarly, dopaminergic transmission also modulates PPI. Dopaminergic receptors are expressed in the amygdala, prefrontal cortex, and striatum [12,13,14,15,16,17]. Administration of dopaminergic agonists into the rat amygdala [18,19], nucleus accumbens [20], or via systemic injection [18,21,22,23] affects PPI. Because dopamine signals stimulus salience to the nucleus accumbens [24], it may be linked to regulating the flow of information to other brain regions by controlling sensorimotor gating [22,25]. Indeed, systemic administration and injection into the rat nucleus accumbens of $D_{1}$/$D_{2}$ or $D_{2}$ receptor agonists, such as amphetamine, apomorphine, and quinpirole, reduce PPI [20,21,26,27,28], while $D_{2}$ antagonists promote an increase [29,30]. Evidence shows a more robust PPI deficit caused by $D_{1}$ receptor agonists administration in mice [31] compared to rats [21,31,32]. The differences in PPI modulation by $D_{1}$ and $D_{2}$ receptors may stem from their differing expression levels in the striatum and the distinct neural pathways within this region [13,33,34].

Pathological conditions, such as Parkinson’s and Huntington’s diseases, [35,36] and psychiatric disorders, as schizophrenia, bipolar, and anxiety disorders [35,37,38,39,40], show deficits in the sensorimotor gating revealed by reduced PPI [41,42], making this test a valuable tool to understand disorders of the nervous system. In particular, a recent meta-analysis showed a significant reduction in PPI of patients with schizophrenia [40]. This psychiatric disorder was associated with functional alterations in the striatum and in the striatal dopaminergic transmission (for a review, see [43]), and with increased amygdala activity and reduced prefrontal cortex metabolism [44,45]. Chronic administration of amphetamine can stimulate dopaminergic system activity and increase dopamine release in the nucleus accumbens [46], partially mimicking the hyperdopaminergia observed in the associative striatum in patients with schizophrenia [47]. These alterations are accompanied by deficits in PPI [46]. In addition, neurodevelopmental animal models of schizophrenia also show activity alterations in the ventral tegmental area, basolateral amygdala, and prefrontal cortex associated with PPI impairment [46,48]. The study of how alterations and dopaminergic transmission in these regions affect PPI could offer insights about the pathophysiology of schizophrenia.

However, it is not yet clear how these modulatory units and the dopaminergic transmission modulate both ASR and PPI. Previous computational models of PPI implemented and simulated brain stem units related to ASR and PPI [49,50,51]. These models reproduced some basic features of PPI detected in animal and human experiments, such as the effect of the variation of the prepulse intensity and the interstimulus interval (ISI) [52,53,54,55]. A recent computational model studied the cellular mechanism for the PPI in the *Drosophila* larvae [56,57], also showing that a similar cellular circuit controls the PPI. However, these models relied mostly on the brainstem units and did not implement neither the units related to the modulation of ASR and PPI nor the dopaminergic transmission. As a result, it was impossible to study the involvement of the modulatory components in these computational models.

In this work, we present a computational model that simulates ASR and PPI by implementing the modulatory units (amygdala, medial prefrontal cortex, nucleus accumbens, ventral pallidum) and dopaminergic transmission (represented by the ventral tegmental area and its connections). Schizophrenic patients present, in general, a significant deficit in PPI and this disease is extensively investigated via pharmacological studies in animal models [40,46,48]. Therefore, in this work, we focused on implementing the modulatory components of PPI that have been associated with schizophrenia to understand how these units interact and impact PPI. The model was used to reproduce basic features of PPI (effect of the variation of prepulse intensity, ISI, and the habituation in ASR) and to simulate the effects of systemic or local (amygdala, ventral pallidum, and nucleus accumbens) GABAergic and dopaminergic drugs. With these tests, we characterized the role of the modulatory units on PPI and ASR.

## 2. Methods

The model includes three different neural pathways, with the first associated with the mediation of ASR, the second associated with the mediation of PPI, and the third associated with the modulation of both ASR and PPI (Figure 1). Each unit in the model represents a different structure of the nervous system that was associated to ASR and PPI in animal studies [5,7,8]. The model input, representing a sound stimulus that reaches the cochlea, is processed through the network, and elicits a motor response represented by the activity of motor neurons in the ASR pathway. The magnitudes of the sound stimuli were chosen to reflect the sound intensities used in experimental settings [58,59]. The model parameters were adjusted to simulate a virtual background with an intensity of 60 dB. Every sound stimuli simulated in this work is considered above a virtual background of 60 dB. Hence, an input I=60 represents a sound with intensity 60 dB above background, i.e., it represents a sound with intensity of 120 dB.

The activity of each unit was modeled by the leaky-integrator model for population activity [50]:(1)$$\begin{array}{c} \frac{dA}{dt}=\frac{1}{\tau }\left[-A+D\cdot GABA\cdot E\cdot I\right], \end{array}$$
where *A* represents the unit activity, τ is the time constant, *D* is the activity of dopaminergic receptors, GABA represents the GABAergic modulation factor (used in drug tests as described in the Section Test Procedures below), *E* and *I* are the excitatory and inhibitory inputs, respectively.

Unless otherwise specified, the excitatory and inhibitory inputs were given by the equations: (2)$$\begin{array}{cc} E & =\sum_{e\in E}\frac{e^{2}}{k_{e}^{2}+e^{2}} \end{array}$$(3)$$\begin{array}{cc} I & =\prod_{i\in I}\left(1-\frac{i^{2}}{k_{i}^{2}+i^{2}}\right), \end{array}$$
where *E* and *I* are, respectively, the sets of all units that make excitatory and inhibitory connections to unit *A*, and $k_{e}$ and $k_{i}$ are semi-saturation constants that control, respectively, the strengths of each excitatory and inhibitory input. Since the excitatory inputs were assumed to be independent of each other (with the exception of the NAc input, as specified in the Appendix A. Model Equations), the total excitatory input to unit *A* was implemented by a sum. In contrast, the total inhibitory input was implemented by a product so that a single inhibitory input, if sufficiently strong, can silence the unit.

Dopaminergic modulation was implemented by $D_{1}$ and $D_{2}$ dopamine receptors, with activities given by the sigmoid functions [60]: (4)$$\begin{array}{cc} D_{1} & =1+\frac{D_{max}}{1+e^{-k_{DA}\left(M_{VTA}+DA-l_{D_{1}}\right)}} \end{array}$$(5)$$\begin{array}{cc} D_{2} & =1-\frac{D_{max}}{1+e^{-k_{DA}\left(M_{VTA}+DA-l_{D_{2}}\right)}}, \end{array}$$
where $D_{max}$ is the maximum effect of dopaminergic receptor activation, $k_{DA}$ is the slope of the sigmoid function, $l_{D_{1}}$ and $l_{D_{2}}$ are, respectively, the semi-saturation constants for $D_{1}$ and $D_{2}$ receptor activations, DA is a factor that simulates the effect of dopaminergic agonists and antagonists (more details in the Test Procedures Section below), and $M_{VTA}$ is the activity of VTA. The units receiving dopaminergic modulation are Amyg, mPFC, NAc, and VP.

In order to implement a threshold function, we used the *p* function defined as follows:(6)$$\begin{array}{c} p\left(x-l\right)=\left\{\begin{array}{c} 1\;\mathrm{if}\;x>l\\ 0\;\mathrm{otherwise}, \end{array}\right. \end{array}$$
where *l* represents the threshold value. This threshold function was implemented in the equations for CPRN, NAc, VTA, phasic released dopamine, and the synaptic weight in the CRN–CPRN connection.

All the model equations and parameters are described in more detail in the Appendix A. Model Equations and Appendix B. Model Parameters.

### 2.1. Test Procedures

A PPI trial is defined as a sequence of stimulus presentations. A pulse alone trial consists of the presentation of a single high intensity sound stimulus (60 dB above background). This trial was denoted as P60. A prepulse + pulse trial consists of the presentation of a prepulse with an intensity of *X* dB above background followed by a pulse of 60 dB. It was denoted by PP*X* + P60. The time interval between the prepulse and the pulse is called the interstimulus interval (ISI). The percentage of inhibition is defined by
(7)$$\begin{array}{c} \%PPI=100\times \frac{max\left(A_{MN,P}\right)-max\left(A_{MN,PP}\right)}{max(A_{MN,P})}, \end{array}$$
where $A_{MN}$ is the motor neuron activity, and $max(A_{MN,P})$ and $max(A_{MN,PP})$ are the maximum values of $A_{MN}$ for pulse trials and prepulse + pulse trials, respectively. Greater %PPI values mean high inhibition, while lower %PPI values indicate low inhibition. We chose to calculate the %PPI to facilitate the comparison with the PPI literature.

In order to test whether the model can reproduce the basic features observed in PPI experiments, we conducted tests varying ISI and prepulse intensity. In the first type of test, the ISI varied from 0 ms to 250 ms in steps of 10 ms between simulations. In this test, the pulse intensity was 60 dB above background and the prepulse intensities were 15, 20, and 25 dB above background. Both stimuli had the duration of 30 ms. In the second type of test, the prepulse intensity varied from 0 dB to 100 dB above background in steps of 5 dB between simulations. The ISIs applied were 60, 70, and 80 ms. The pulse intensity was 60 dB above background and both stimuli had the duration of 30 ms. We calculated the %PPI for each ISI and prepulse intensity in these tests. We chose to test PPI with the above indicated sound intensity above background and ISI values to avoid floor and ceiling effects, and because they are commonly used in PPI experiments [28,52,53,55,61,62].

A PPI session was defined as a sequence of trials of pulse alone, prepulse alone, prepulse + pulse or no stimulus repeated by eight times for each trial type separated by a random intertrial interval (ITI). To verify if the model can produce ASR habituation by repeated pulse presentation, we carried a test with 10 repeated P60 trials. In this test, the pulse intensity was 60 dB above background with duration of 30 ms and the ITI was randomly drawn from an uniform distribution of integers in the intervals [5 s, 15 s], [10 s, 15 s], and [20 s, 25 s].

In order to simulate different conditions, mimicking an experiment with a group of different animals, the PPI session tests (habituation test, GABAergic test, and dopaminergic test) were simulated with variability in the model parameters. In each simulation, the values of all model parameters were randomly drawn from uniform distributions in the intervals centered at the parameter values in Table A1 with widths of 10% of those values in each side. This process was repeated 10 times for each parameter at each new simulation, resulting in 10 different sets of parameters at each simulation to mimic an experiment with 10 different animals. For example, the 10 values of parameter $k_{I}$ at each new simulation were drawn from the uniform distribution in the interval [31.5 dB, 38.5 dB].

### 2.2. Drug Action Tests

In the tests with the simulation of GABAergic drugs, we used values of the GABA factor smaller than 1 (0.00≤GABA<1.00) to simulate a GABAergic agonist drug, which reduced the activity of the unit under test. In contrast, to simulate the effect of a GABAergic antagonist, we used values of GABA greater than 1 (1.00<GABA≤2.00), which increase the unit activity. The control case was defined as GABA=1.00. The maximum inhibition of the unit occurs for GABA=0.00 and the maximum hyperactivation for GABA=2.00.

To test the interaction between Amyg and VP in a GABAergic test, both the GABA factors in Amyg (Equation (Equation 18)) and VP (Equation (Equation 28)) were varied from 0.00 (maximum inhibition) to 2.00 (maximum hyperactivation) in steps of 0.10 between the simulations. In this test, we simulated prepulse + pulse trials of PPI with prepulse intensity of 25 dB above background, pulse of 60 dB above background, ISI of 80 ms, and duration of both stimuli of 30 ms.

We conducted a test simulating a PPI session with GABAergic manipulation in Amyg and VP. The session started with the presentation of 10 pulse trials (60 dB above background, 30 ms) followed by a pseudorandom presentation of 64 trials of pulse, prepulse (15, 20, or 25 dB above background, 30 ms), prepulse + pulse (ISI of 80 ms), or no stimulus. Each trial type was repeated eight times along the session. The ITI was 10–15 s. In this test we analyzed both ASR for the pulse and each prepulse + pulse intensity and the %PPI for each prepulse intensity. To mimic an experiment conducted with ten different animals, the session was simulated ten times with different values of model parameters (as described above). This test had the following groups: Control (GABA=1.00 in Amyg and VP), Amyg group (GABA=0.20 in Amyg), VP group (GABA=0.20 in VP), and Amyg-VP group (GABA=0.20 in Amyg and VP). The average ASR and %PPI between the groups were compared with a two-way repeated measure ANOVA with Trat1 (Control × GABA Amyg) and Trat2 (Control × GABA VP) as between subject factor and intensity (15, 20, or 25 dB above background) as within subject factor. The level of significance was *p* < 0.050. We performed the statistical analysis using the open platform JAMOVI (release 2.5.2). Tests for normality were made with the Shapiro–Wilk test (see Appendix C) and for homoscedasticity with Mauchly’s sphericity test [63].

The effects of dopaminergic drugs were modeled by using −1.00≤DA<0.00 to simulate the effect of a dopaminergic antagonist, so that the activity of the dopaminergic receptor is reduced; on the other hand, the effect of a dopaminergic agonist was simulated by using 0.00<DA≤1.00, which increased the activity of the dopaminergic receptor. The control condition was defined as DA=0.00. The maximum inhibition of the dopaminergic receptor occurs with DA=−1.00 and the maximum hyperactivation, with DA=1.00. In the systemic simulation test, all dopaminergic receptors in the model had their DA factor manipulated. Whereas in the Amyg simulation test, only the dopaminergic receptors in the Amyg unit were manipulated (Equations (Equation 17) and (Equation 18)). In the NAc simulation test, only the dopaminergic receptors in NAc were manipulated (Equations (A15) and (A17)). When simulating drugs with action on both $D_{1}$ and $D_{2}$ receptors, the DA factor on both receptors was manipulated. For drugs with action on only one of the receptors, only the DA factor of the respective receptor was manipulated.

We also conducted a test simulating a whole PPI session with manipulation of the DA factor. The session was the same as the session in the GABAergic test and we simulated 10 different parameter sets (representing 10 different animals). The dopaminergic session test had the following groups: Control (DA factor in all receptors set to 0.00), Systemic group (DA factor in all receptors set to 0.50), Amyg group (DA factor of $D_{1}$ and $D_{2}$ receptors in Amyg set to 0.50), and NAc group (DA factor of $D_{1}$ and $D_{2}$ receptors in NAc set to 0.50). The average ASR and %PPI between the groups were analyzed with a one-way repeated measure ANOVA with Group (Control × Systemic × Amyg × NAc) as between subject factor and intensity (15, 20, or 25 dB above background) as within subject factor. The level of significance was *p*
<0.050.

### 2.3. Computational Methods

The system of differential equations was numerically integrated using the Euler method with time step dt=0.02 ms. The numerical routines were implemented in Python 3.11.5 using the packages NumPy (1.24.3) and Numba (0.57.1) (for JIT compilation) and the graphs were built with Matplotlib (3.7.2).

## 3. Results

A PPI trial was simulated by the presentation of a prepulse with intensity of 25 dB above background, a pulse with intensity of 60 dB above background, ISI of 80 ms, and duration of both stimuli of 30 ms. The sound stimuli presentation increased every unit activity (Figure 2). In the ASR pathway, both Ch and CRN present two peaks of activity, the first one corresponding to the prepulse presentation and the second one to the pulse. However, in CPRN, due to the transfer threshold and the inhibitory input from PPTg, there is only one peak of activity corresponding to the pulse with smaller amplitude when compared to the peaks in Ch and CRN. The same is observed in the MN. That reduction in the activity amplitude in the response for the pulse is what characterizes the PPI. A similar response is observed in the PPI pathway. Both IC and SC show two peaks of activity. The PPTg shows one peak with high amplitude corresponding to the prepulse and after that a lower activity caused by the pulse. The activities of the ASR and PPI pathways also propagate to the modulatory pathway. In the units representing the dopaminergic transmission in NAc, it is possible to note slight increases in the extracellular dopamine concentration and in the activity of presynaptic $D_{2}$ receptors. The excitatory and inhibitory subpopulations in the mPFC and in the amygdala showed two peak responses, except for the excitatory subpopulation in the amygdala. Since the amygdala’s excitatory subpopulation received a direct inhibitory input from the inhibitory subpopulation, the peak response amplitude for the pulse was attenuated in the excitatory subpopulation. In the same way, both direct and indirect NAc pathways responded for the stimuli. However, the response for the pulse in the NAcI was also attenuated. Since the VP only receives an inhibitory input from NAcI, its activity was reduced by the stimuli presentation. Finally, the VTA also showed a peak response for the prepulse while the activity in response for the pulse was attenuated.

### 3.1. Basic Features of PPI

In order to verify if the present model would reproduce some basic features observed in animal experiments of PPI, we conducted tests varying the duration of ISI, the prepulse intensity, and verified the habituation of ASR by repeated pulse presentations.

#### 3.1.1. ISI Test

We simulated trials of prepulse + pulse in which the ISI varied from 0 ms to 250 ms in steps of 10 ms between the simulations. The pulse intensity was fixed at 60 dB above background and the prepulse intensities tested were 15, 20, and 25 dB above background. The duration of pulse and prepulse were 30 ms.

The results show the %PPI are dependent on the ISI variation (Figure 3A). The %PPI first increase as the ISI rises and reaches its maximum value with ISI between 70 and 100 ms. For a prepulse of 15 dB above background, the maximum %PPI was 88.59% with ISI = 90 ms; for a prepulse of 20 dB, the maximum %PPI was 85.70% with ISI = 80 ms and for a prepulse of 25 dB, the maximum %PPI was 84.82% with ISI = 80 ms. Above these values of ISI, the %PPI decreases to a baseline around 0% of inhibition. For ISIs below 50 ms, the %PPIs obtained in the simulations were negative. For an ISI of 30 ms, there was a %PPI of −9.06% for the prepulse with intensity of 15 dB, a %PPI of −14.15% for the prepulse with 20 dB and a %PPI of −18.89% for the 25 dB prepulse.

#### 3.1.2. Prepulse Intensity Test

To examine the relationship between %PPI and the intensity of the prepulse, we simulated trials of prepulse + pulse in which the intensity of prepulse varied from 0 dB to 100 dB above the background in steps of 5 dB between the simulations. The ISI was set to 60, 70, or 80 ms and the duration of both pulse and prepulse to 30 ms.

This test shows that the %PPI depends also on the prepulse intensity (Figure 3B). For prepulse intensities from 0 dB to 20 dB above background, the %PPI increases, reaching its maximum with prepulse intensities between 20 dB to 40 dB above background. For the ISI of 60 ms, the maximum inhibition was 55.35% for a prepulse of 40 dB; for ISI of 70 ms, the maximum %PPI was 82.02% with the prepulse of 35 dB and for the ISI of 80 ms 85.70% for a prepulse of 20 dB. For prepulses with intensities above 40 dB, the %PPI decreases, reaching negative values for prepulse intensities greater than 60 dB for all ISI. For a prepulse with intensity of 80 dB, the %PPI obtained was −11.00% for the three ISIs. For a prepulse of 100 dB, the %PPI was −14.92%, independetly of the ISI value.

For the remaining tests, the stimuli parameters were set according to the results shown in this section: the ISI was fixed in 80 ms, the prepulse intensity was 25 dB above background, the pulse was fixed in 60 dB above background, and the duration of both prepulse and pulse fixed at 30 ms. Simulating trials of PPI with these values led to a %PPI of 84.82%. We selected these parameters in order to avoid ceiling effect and reflect experimental settings.

#### 3.1.3. ASR Habituation Test

The PPI session was composed of 10 pulse trials and ITIs of 5–15 s, 10–15 s, and 20–25 s. For each ITI, we ran 10 simulations with random initial parameter values, as described above. Repeated presentation of pulses decreased the MN activity amplitude (Figure 4) for all ITI used. For each pulse presentation, there was a decrease in MN activity, from 0.612 (in all ITIs) to 0.543 (ITI 5–15 s, Figure 4A), 0.561 (ITI 10–15 s, Figure 4B) and 0.595 (ITI 20–25 s, Figure 4C). The percentage of decrease in MN activity was calculated in reference to the activity in response to the first pulse (Figure 4D). The simulation showed that the percentage of decrease in MN activity depends on the ITI used in the test. After 5 pulse presentations, the MN activity decreased by 10.51% for the ITI of 5–15 s, by 6.87% for the ITI of 10–15 s, and by 2.53% for the ITI of 20–25 s. After 10 pulse presentations, the MN activity decreased by 11.14% for ITI of 5–15 s, by 8.31% for the ITI of 10–15 s, and by 2.79% for the ITI of 20–25 s. Thus, the highest habituation was observed for the shortest ITI used (5–15 s).

### 3.2. Manipulation of GABAergic Transmission

#### 3.2.1. GABAergic Test: Trial

To test the influence of the amygdala and ventral pallidum on PPI and simulate the effect of GABAergic drugs, we manipulated the GABA factor in only the amygdala (Equations (Equation 17) and (Equation 18)), only in the VP (Equation (Equation 28)), or in both units. The GABA factor varied from 0.0 (highest inhibition, simulating a GABAergic agonist) to 2.0 (highest hyperactivation, simulating a GABAergic antagonist) in steps of 0.10 between the simulations (mode details in Section 2.2). In this test, the trials had a prepulse + pulse with pulse intensity of 60 dB above background, prepulse intensity of 25 dB, ISI of 80 ms, and duration of prepulse and pulse of 30 ms.

Affecting amygdala activity resulted in a decrease in %PPI, for both inhibition (GABA factor below 1.00) and hyperactivation (GABA factor above 1.00, Figure 5) of this unit. The simulation with GABA factor at 0.50 had a %PPI of 75.48% (reduction of 9.34% compared to control) and with GABA factor at 0.00 (maximum inhibition) had a %PPI of 59.81% (reduction of 25.01% compared to control). The simulation of GABA factor at 1.50 (hyperactivation of amygdala) had a %PPI of 63.30% (reduction of 21.52% compared to control) and with GABA factor at 2.00 had %PPI of 58.77% (reduction of 26.05% compared to control).

Manipulating only the GABA factor in VP reduced the %PPI only for the inhibition (GABA factor lower than 1.00, Figure 5). The simulation with GABA factor at 0.50 had %PPI of 75.60% (reduction of 9.22% compared to control) and GABA factor at 0.00 (maximum inhibition) had a %PPI of 68.23% (reduction of 16.59% compared to control). However, for GABA factor above 1.00, there was only a little difference from control. For GABA factor of 1.50, the simulation had a %PPI of 85.70% (increase of 0.88% compared to control) and for GABA factor at 2.00 had a %PPI of 83.43% (reduction of 1.39% compared to control).

The reduction in %PPI caused by amygdala inhibition was further increased by the hyperactivation of VP (Figure 5). The simulation with GABA factor at 0.50 in amygdala and 1.50 in VP had a %PPI of 36.25% (reduction of 48.56% compared to control). With GABA factor at 0.00 in amygdala and 2.00 in VP, the simulation resulted in a %PPI of 19.38% (reduction of 65.44% compared to control).

A similar reduction was observed by hyperactivating amygdala and inhibiting VP (Figure 5). The simulation with GABA factor in amygdala at 1.50 and in VP at 0.50 had a %PPI of 56.07% (reduction of 28.75% compared to control) and the simulation with GABA factor in amygdala at 2.00 and in VP at 0.00 had a %PPI of 49.29% (reduction of 35.53% compared to control).

Likewise, hyperactivating both amygdala and VP reduced the %PPI in the simulation (Figure 5). For GABA factor at 1.50 in both amygdala and VP, the %PPI simulated was 71.34% (reduction of 13.48% compared to control) and at 2.00, the %PPI was 74.08% (reduction of 10.73% compared to control).

Finally, inhibition of both amygdala and VP prevented the reduction in the %PPI caused by the inhibition of only amygdala or only VP (Figure 5). The simulation with GABA factor at 0.50 in both amygdala and VP resulted in a %PPI of 87.20% (increase of 2.32% compared to control), a %PPI 8.62% higher than the condition with inhibition in amygdala (GABA factor of 0.50 in amygdala), and 8.73% higher than the condition with inhibition of VP (GABA factor of 0.50 in VP). The simulation with GABA factor at 0.00 in amygdala and VP had a %PPI of 84.22% (reduction of 0.60% compared to control), 24.41% higher than %PPI for the condition with GABA factor at 0.00 in amygdala, and 15.99% higher than the condition with GABA factor at 0.00 in VP.

#### 3.2.2. GABAergic Test: Session

In order to test the effect of simulating GABAergic agonist during a PPI session and if the effects of the manipulation of GABA factor could be attributed to alterations in the MN activity (ASR), we conduct a session test composed of 74 trials (as described in Section 2.1) including pulse (60 dB above background), prepulses (intensities of 15, 20, and 25 dB above background), prepulse + pulse, and no stimuli. This test session was repeated ten times varying the model’s initial parameters. The session was conducted with four groups: the VP group (GABA factor at 0.20 in VP), the Amygdala group (GABA factor at 0.20 in amygdala), Amygdala-VP group (GABA factor at 0.20 in both VP and amygdala), and Control group (GABA factor at 1.00 in both units). This process mimics an experiment with ten different animals in each treatment group with the administration of muscimol (GABAergic agonist) intra-amygdala, intra-VP, or in both regions. The data obtained in this test were analyzed with a two-way repeated measure ANOVA, with intensity as within subject factor and Treat1 (Control × GABA factor in amygdala at 0.20) and Treat2 (Control × GABA factor in VP at 0.20) as between subject factors. Since the data violated homoscedasticity (Mauchly’s sphericity test; ASR W = 0.133, p<0.001, Greenhouse-Geisser ε = 0.622; %PPI W = 0.219, p<0.001, Greenhouse-Geisser ε = 0.561), we applied the ANOVA test with Greenhouse-Geisser correction [63]. Both ASR and %PPI are normally distributed (the results of the Shapiro–Wilk test can be found in Table A2 in Appendix C).

The ANOVA showed a main effect of intensity [F(1.87, 67.22) = 2674.2, *p* < 0.001], Treat1 [F(1, 36) = 10.3, *p* < 0.01] and Treat2 [F(1, 36) = 12.3, *p* < 0.001], and a significant interaction Treat1 × Treat2 [F(1, 36) = 28.3, *p* < 0.01] (Figure 6A). The Tukey *Post-hoc* test showed that the ASR for the pulse was higher than the ASR for the prepulse + pulse for every prepulse intensity [*p* < 0.001]. There was no difference in the ASR to P60 between the four groups. However, the Amygdala group showed a higher amplitude for the PP15 + P60 and PP20 + P60 compared to Control group [*p* < 0.001], VP group [*p* < 0.001], and Amygdala-VP group [*p* < 0.001]. The amygdala group showed a higher ASR amplitude for the PP25 + P60 just compared to Control group [*p* < 0.001] and to Amygdala-VP group [*p* < 0.001].

The two-way repeated measure ANOVA for the %PPI revealed a main effect of Treat1 [F(1, 36) = 26.8, *p* < 0.001] and Treat2 [F(1, 36) = 32.7, *p* < 0.001] and a significant interaction Treat1 × Treat2 [F(1, 36) = 74.3, *p* < 0.001] (Figure 6B). There was no effect of intensity in the %PPI [F(1.12, 40.42) = 2.26, *p* = 0.139]. The Tukey *Post-hoc* test showed that the Amygdala group had a lower %PPI compared to control group [*p* < 0.001] and to the Amygdala-VP group [*p* < 0.001]. However, the Amygdala-VP group did not differ from the control group [*p* = 0.088]. The Amygdala group showed a lower %PPI for PP15 and PP20 compared to the Control group [*p* < 0.001], VP group [*p* < 0.001], and Amygdala-VP group [*p* < 0.001]. The %PPI for the PP25 was lower in Amygdala group compared to Control group [*p* < 0.001] and to Amygdala-VP group [*p* < 0.001]. The Amygdala-VP group showed a higher %PPI for PP25 compared just to the VP group [*p* < 0.001] and the VP group had a lower %PPI for PP25 compared to the Control group [*p* < 0.010].

### 3.3. Manipulation of Dopaminergic Transmission

#### 3.3.1. Dopaminergic Test: Trial

We tested the effect of dopaminergic transmission in the model manipulating the DA factor in the equations which represent the dopaminergic receptors (Equations (4) and (5)) in all units of the model (systemic test), only in amygdala (amygdala test), or only in NAc (NAc test). To distinguish the manipulation over the $D_{1}$ and $D_{2}$ subtypes of dopaminergic receptor, only the DA factor in the equations representing the subtype of receptor under test was manipulated. Thus, in the test of $D_{1}$ receptors, only in these receptors the DA factor varied. In the test of $D_{1}$ and $D_{2}$ receptors, both types of receptors had the DA factor manipulated. The DA factor varied from −1.00 (simulating an antagonist, inhibiting the receptor activity) to 1.00 (simulating an antagonist, activating the receptor) in steps of 0.10 between the simulations. The trials had a pulse intensity of 60 dB above background, prepulse intensity of 25 dB, ISI of 80 ms, and stimuli duration of 30 ms.

#### 3.3.2. Systemic Test

In the systemic test with manipulation of both receptors types, there was a reduction in the %PPI by the increase in the DA factor (simulating an agonist) and a slight increase in %PPI by the reduction of DA factor (simulating an antagonist) (Figure 7). For the DA factor at 0.50 the simulation had a %PPI of 19.23% (reduction of 65.59% compared to control) and at 1.00 the simulation had a %PPI of 13.71% (reduction of 71.11% compared to control). In contrast, the simulation with the DA factor at −0.50 had a %PPI of 89.13% (increase of 4.31% compared to control) and with the DA factor at −1.00 had a %PPI of 89.12% (increase of 4.30% compared to control).

Similarly, in the systemic test with manipulation of only $D_{1}$ receptors, the simulation of increased DA factor caused a decrease in the %PPI, whereas reducing the factor’s value led to an increase in the %PPI (Figure 7). For the DA factor at 0.50 the simulation led to a %PPI of 59.84% (reduction of 24.98% compared to control) and at 1.00 to a %PPI of 53.77% (reduction of 31.05% compared to control). For the DA factor at −0.50 the %PPI was 88.66% (increase of 3.85% compared to control) and at −1.00 the %PPI was 89.13% (increase of 4.31% compared to control).

Increasing the DA factor only in $D_{2}$ receptors caused a reduction in the %PPI while the reduction of the DA factor caused an increase in the %PPI (Figure 7). For the DA factor at 0.50 the %PPI was 36.70% (reduction of 48.12% compared to control) and at 1.00 the %PPI was 21.41% (reduction of 63.41% compared to control). In contrast, the simulation with DA factor at −0.50 had a %PPI of 89.17% (increase of 4.35% compared to control) and the factor DA at −1.00 had a %PPI of 89.13% (increase of 4.31% compared to control).

#### 3.3.3. Amygdala Test

In the amygdala test, increasing the value of DA factor in both receptor types led to a reduction in the %PPI, while reducing the DA factor in both receptors caused an increase in the %PPI (Figure 7A). The simulation with the DA factor at 0.50 led to a %PPI of 58.21% (reduction of 26.60% compared to control) and for the factor DA at 1.00 the %PPI was of 55.11% (reduction of 29.71%). In contrast, for the factor DA at −0.50 the %PPI was 88.35% (increase of 3.53% compared to control) and at −1.00 the %PPI was 88.25% (increase of 3.44% compared to control).

The amygdala tests with manipulation of only $D_{1}$ receptors (Figure 7A), when DA factor was set at 0.50, %PPI was 62.66% (decrease of 22.16% compared to control) and at 1.00 had a %PPI of 58.16% (decrease of 26.66% compared to control). The increased DA factor in $D_{1}$ receptors in amygdala also caused an increase in the activity of excitatory subpopulation and a decrease in the inhibitory subpopulation (Figure 7B,C). For DA factor at −0.50 the simulated %PPI was 88.29% (increase of 3.47% compared to control) and at −1.00 the %PPI was 89.06% (increase of 4.25%). This was accompanied by a decrease in excitatory amygdala subpopulation activity and a slight increase in the inhibitory subpopulation activity (Figure 7B,C).

The amygdala tests with manipulation of only $D_{2}$ receptors (Figure 7A), when DA factor was set at 0.50, %PPI was 65.63% (decrease of 19.18% compared to control) and at 1.00, %PPI was 65.25% (decrease of 19.57% compared to control). The increase in DA factor in the $D_{2}$ receptor in amygdala caused an increase in the excitatory subpopulation activity but a slight decrease in the inhibitory subpopulation activity (Figure 7B,C). The simulation with DA factor at −0.50 led to a %PPI of 88.49% (increase of 3.67% compared to control) and at −1.00 to a %PPI of 88.49% (increase of 3.67% compared to control, Figure 7A). This manipulation was associated with a decrease in the activity of excitatory subpopulation and a slight increase of inhibitory subpopulation (Figure 7B,C).

#### 3.3.4. NAc Test

The NAc tests with manipulation of DA factor for both receptors or only $D_{2}$ receptor affected the %PPI. The manipulation of DA factor only in $D_{1}$ receptors led to a slight change in the %PPI compared to the manipulation of $D_{2}$ receptors or both types receptors (Figure 7A). The simulation with the DA factor of both $D_{1}$ and $D_{2}$ receptors in NAc at 0.50 led to a %PPI of 62.89% (reduction of 21.93% compared to control) and at 1.00 to a %PPI of 38.48% (reduction of 46.34% compared to control). This manipulation of DA factor in NAc was accompanied by a slight increase in NAc activity for an intermediate value of the factor DA (at 0.2, Figure 7D) and a reduction in NAc activity for high values of DA factor (Figure 7D). In contrast, the intermediate value of DA factor in NAc caused a slight increase in VP activity while higher values also led to higher increases in VP activity (Figure 7E). For the DA factor at −0.50 the %PPI obtained was 89.80% (increase of 4.98% compared to control) and at −1.00 the %PPI was 90.21% (increase of 5.39% compared to control). This manipulation caused a decrease in NAc activity and an increase in VP activity (Figure 7D,E).

For the manipulation of only $D_{2}$ receptors in NAc (Figure 7A), the simulation with the DA factor at 0.50 had a %PPI of 68.08% (reduction of 16.74% compared to control) and at 1.00 had a %PPI of 52.58% (reduction of 32.24% compared to control). Similar to the manipulation in both receptors, the increase in the DA factor of $D_{2}$ receptors in NAc caused in NAc activity a slight increase for intermediate values (at 0.2) and a decrease for higher values (Figure 7D). There was an increase in VP activity for the increased DA factor in NAc $D_{2}$ receptors (Figure 7E). For DA factor at −0.50, the simulated %PPI was 89.56% (increase of 4.74% compared to control) and at −1.00 the %PPI was 90.10% (increase of 5.28% compared to control, Figure 7A). The reduction of DA factor in NAc led to a reduction in NAc activity and an increase in VP activity (Figure 7D,E).

In contrast, for the manipulation of only D_1_ receptors in NAc (Figure 7A), for the DA factor at 0.50 the %PPI was 84.92% (increase of 0.10% compared to control) and at 1.00, the %PPI was 85.16% (increase of 0.34% compared to control). This manipulation was associated with a slight increase in NAc activity (Figure 7D). For the DA factor at −0.50, the %PPI was 86.50% (increase of 1.68% compared to control) and at −1.00 the %PPI was 86.76% (increase of 1.94% compared to control). The reduction in DA factor of $D_{1}$ receptors in NAc led to a reduction of NAc activity (Figure 7D) and an increase in VP activity (Figure 7E).

#### 3.3.5. Dopaminergic Test: Session

The PPI session was composed of 74 stimuli presentation including pulse (60 dB above background), prepulse (intensities of 15, 20, and 25 dB above background), prepulse + pulse, and background. This test was repeated ten times varying the model’s parameters. The tests groups were as follows: Systemic group (DA factor at 0.50 in both $D_{1}$ and $D_{2}$ receptors and all receptors in every unit of the model), Amygdala group (DA factor at 0.50 in both $D_{1}$ and $D_{2}$ receptors in amygdala), NAc group (DA factor at 0.50 in both $D_{1}$ and $D_{2}$ receptors in NAc), and Control group (DA factor at 0.00 in all receptors). The data was analyzed with an one-way repeated measure ANOVA, with intensity as within subject factor and Group as between subject factor. Since the ASR and %PPI obtained in this test violated homoscedasticity (Mauchly’s sphericity test; ASR W = 0.093, p<0.001, Greenhouse-Geisser ε = 0.589; %PPI W = 0.230, p<0.001, Greenhouse-Geisser ε = 0.565), we conducted the ANOVA test with Greenhouse-Geisser correction [63]. Both ASR and %PPI are normally distributed (the results of Shapiro-Wilk test are shown in Table A3 in Appendix C).

For the ASR (MN activity in the present model), the ANOVA showed a main effect for intensity [F(1.77, 63.57) = 1060.40, *p* < 0.001] and Group [F(3, 36) = 37.9, *p* < 0.001] (Figure 8A). The Tukey *post-hoc* test revealed that the amplitude of ASR for the pulse was higher than any ASR for prepulse + pulse [*p* < 0.001]. There was no difference between the groups in the ASR amplitude in response to P60. The ASR for PP15 + P60 in the Systemic group was higher than the Control group [*p* < 0.001] and Amygdala group [*p* < 0.001], and the ASR for PP15 + P60 in the NAc group was higher than the Control [*p* < 0.001] group and Amygdala group [*p* < 0.001] but lower than the Systemic group [*p* < 0.001]. The ASR for PP20 + P60 in the Systemic group was higher than the Control group [*p* < 0.001], Amygdala group [*p* < 0.001] and NAc group [*p* < 0.001], in Amygdala group was higher than the Control group [*p* < 0.001], and in NAc was higher than Control group [*p* < 0.001]. The ASR for PP25 + P60 in the Systemic group was higher than the Control group [*p* < 0.001], Amygdala group [*p* < 0.001], and NAc group [*p* < 0.001], in Amygdala group was higher than the Control group [*p* < 0.001], and in NAc group was higher than Control group [*p* < 0.001].

The ANOVA also showed a main effect of intensity [F(1.13, 40.69) = 3.95, *p* = 0.049 < 0.05] and Group [F(3, 36) = 89.5, *p* < 0.001] for the %PPI (Figure 8B). The Tukey *post-hoc* test showed that the %PPI for the 15 dB prepulse was lower than the %PPI for the 20 dB prepulse [*p* < 0.050] and that the control group had higher %PPI than any other group [*p* < 0.001]. The %PPIs for PP15 in Systemic group and in NAc group were lower than the Control group [*p* < 0.001], and both Amygdala group and NAc group had higher %PPIs than the Systemic group [*p* < 0.001], and NAc group showed a lower %PPI than the control group [*p* < 0.001]. The Systemic group, Amygdala group, and NAc group showed lower %PPIs than the Control group [*p* < 0.001] for PP20. Both Amygdala and NAc groups showed higher %PPIs for PP20 than the Systemic group [*p* < 0.001]. All groups showed lower %PPIs for PP25 compared to Control group [*p* < 0.001]. Again, the Amygdala and NAc groups showed higher %PPIs than the Systemic group [*p* < 0.001].

## 4. Discussion

The model developed in the present work reflects the brain regions involved in the mediation of ASR and PPI as shown by the electrophysiological, pharmacological, and lesion reports [1,3,4,5,6]. The model developed showed that %PPI depends on the ISI and the prepulse intensity. It also demonstrated the habituation in the ASR by repeated pulse presentation. These three features are observed in experiments of PPI in animals and humans. Furthermore, ASR and PPI modulatory units were implemented in the present model, allowing us to simulate how these units regulate PPI and ASR pathways and to evaluate the effects of GABAergic and dopaminergic drugs. The simulation of GABAergic agonists and antagonists caused a decrease in the %PPI without affecting the ASR for the pulse. Likewise, the simulation of dopaminergic agonists produced a decrease in the %PPI, impacted the amygdala, NAc and VP activities, but did not change the ASR to pulse. In contrast, the simulation of dopaminergic antagonists led to an increase in %PPI, also modulating the activity of amygdala, NAc, and VP.

The startle response can be mediated by a circuit connecting a sensory input (like the Ch for auditory inputs in ASR, but also for vision and tactile inputs from different pathways) with a motor response unit. In rats, the ASR pathway is composed by the Ch, CRN, CPRN, and MN [5]. Similarly, the PPI can be mediated by a parallel circuit starting from a sensory system that terminates at a nucleus that inhibits the ASR pathway. In the rat brain, this circuit is composed by the SC, IC, and PPTg, the last nucleus inhibiting the CPRN [5,6]. However, other animals also show the startle modulation by the sensorimotor gating and some have a similar circuit organization involving a pathway connecting sensory receptors with a motor system, like the larvae of Drosophila melanogaster and the molusk Tritonia diomedea [64,65]. This fundamental mechanism indicates that sensorimotor gating remains conserved across the animals and similar circuits mediate this function [2]. Thus, the conditions and parameters, such as the ISI and prepulse intensity, for which we detect an inhibition of the startle response, reflects the functioning and connections within this brain circuit.

The intervals obtained in the present study for the ISI and prepulse intensity for which there was a maximum %PPI are consistent with the experimental reports. In rats, the maximum %PPI is obtained for an ISI between 20 ms and 100 ms and for prepulses with 15 to 20 dB above the background [52,53,54,55]. The interval between the prepulse and the pulse for the inhibition of ASR reflects the circuits and the parallel mechanism involved in the mediation of PPI. Since the PPI pathway has an inhibitory output to the ASR pathway, the ISI should last enough for the output of the PPI pathway to inhibit the ASR pathway activated by the pulse. Similarly, the prepulse intensity should be able to activate the PPI pathway without reaching the threshold to activate the ASR pathway. Hence, there are optimal intervals for the ISI and prepulse intensity for which the PPI is observed.

The habituation in startle response also reflects the functioning of the ASR pathway. In rats, the habituation of ASR is mediated by the STD mechanism in the connection between the CRN and CPRN [66]. The tests conducted with the present computational model showed that a higher habituation is achieved with shorter ITIs. As observed in rats, the percentage of decrease in ASR amplitude is also greater for shorter ITI [52,53,54,55]. Since the habituation mechanism depends mainly on the STD in the CRN-CPRN connection, the observation that shorter ITIs leads to higher decreases in ASR is consistent with the proposed mechanism.

The units chosen to compose the modulatory pathway reflect the pharmacological studies [5] and reports with neuroimaging techniques [7,8]. Rohlender and colleagues [7,8] using PET scan with the metabolic tracer [^18^F]fluoro-2-deoxyglucose in rats showed that the amygdala, mPFC, NAc, VP, and VTA are indeed involved on the PPI modulation. In that study, they found that the VTA activation is negatively correlated with the PPI and that NAc and basolateral amygdala are positively correlated with PPI. They also showed that the modulatory regions could be divided into a lateral system, related to the emotional processing of stimuli, and a medial one, involved in the cognitive processing. The VTA and the left prelimbic PFC (part of the mPFC) are part of the medial system. The amygdala, PPTg, and right prelimbic PFC are part of the lateral system. Since the sensorimotor gating acts protecting the interference of information processing by a stimulus that evokes the ASR [67], the modulatory units regulate the efficacy of the prepulse to inhibit the pulse ASR. Thus, while the lateral system is related to the regulation of prepulse inhibition tonus, the medial system regulates the ASR, increasing the response.

The present model offers a mechanistic way to interpret the effect of modulatory regions (amygdala, NAc, and VP) and dopaminergic transmission over the PPI. Simulation of amygdala and VP inhibition resulted in a reduction of %PPI, which is consistent with findings in rats showing that the administration of muscimol into amygdala [9] and the lesion of VP [68] reduced %PPI. However, in the simulations with weaker inhibition of VP, there was no %PPI reduction, which is consistent with another report showing no effect over the %PPI when VP is inactivated by muscimol local administration in rats [9]. In contrast, the simultaneous inhibition of amygdala and VP attenuated the reduction in the %PPI caused by the inhibition of each unit. This result can be interpreted considering the amygdala-NAc-VP pathway. Since amygdala increases NAc activity and VP inhibits PPTg, the reduction of amygdala activity also reduces NAc inhibition over VP, which in turn increases the inhibition over PPTg, leading to reductions in %PPI. By inhibiting both amygdala and VP, the effect of this last unit over PPTg is attenuated. It is noteworthy that this result follows the work of Forcelli et al. [9] showing that the co-administration of muscimol intra-amygdala and intra-VP reverted the PPI deficits caused by the injection of muscimol intra-amygdala.

The dopaminergic modulation of PPI was implemented in the present model by adding a unit representing the VTA and its connections with amygdala, mPFC, NAc, and VP. The simulation of dopaminergic agonists in the systemic test, in amygdala, or in NAc caused a decrease in the %PPI. Similar results are shown in studies with the administration of dopaminergic drugs in rats by systemic injection [22,23,62], intra-amygdala [18], or intra-NAc [20,26,27]. In contrast, the simulation of the dopaminergic antagonists produced a slight increase in the %PPI. This result is also consistent with the literature, showing that the administration of dopaminergic antagonists in rats also slightly increases the %PPI [29,30].

The effect of the dopaminergic modulation over the PPI can be interpreted considering the effects of dopamine in each unit. In the amygdala, the inhibitory interneurons express the $D_{2}$ receptors, while the projection neurons express the $D_{1}$ receptors [14]. The simulation of dopaminergic agonists in the amygdala increases its activity and, as shown previously, the increased amygdala activity has as a result the inhibition of PPTg, causing the reduction in the %PPI. However, our simulation of dopaminergic antagonists in amygdala caused a modest increase in the %PPI. A study conducted by Stevenson and Gratton [19] detected a reduction in the %PPI by injecting raclopride (a $D_{2}$ antagonist) in a rat’s amygdala. As detailed above, the amygdala model implemented in the present work was based on rat studies [14] with different strains (Wistar and Sprague–Dawley) [18,22], while the the study of Stevenson and Gratton [19] used the Long–Evans strain. Since there are differences in the effect of amphetamine between Sprague–Dawley and Long–Evans [26], the difference detected in the simulation of $D_{2}$ antagonist with the study of Stevenson and Gratton could be attributed to the distinct strains used to implement the present model.

As shown by Rohlender and colleagues [7,8], the amygdala is part of the lateral system, regulating the emotional processing and the PPI efficacy. The amygdala signalizes the context aversiveness, increasing or decreasing the PPI tonus. Thus, the increased amygdala activity (by GABAergic antagonist or dopaminergic agonists) could signalize an aversive context, increasing the animal responsivity to environmental stimuli, leading to deficits in PPI. Similarly, since amygdala regulates the PPI tonus, by decreasing its activity simulating the GABAergic agonist, it could interfere with the prepulse efficacy to inhibit the pulse ASR, also impairing the PPI. Thus, the amygdala shows a bimodal effect over the PPI modulation, either hyperactivating or inhibiting the impact of the prepulse.

The dopaminergic transmission also modulates NAc activity. For lower values of DA factor in NAc (representing a lower concentration of dopamine or dopaminergic drugs), there was an increase in its activity, but for higher values of DA factor in NAc, there was a decrease in the activity. Some studies found in animals a decrease in NAc activity caused by dopamine [69,70] while others found an increased activity [71]. In the present model, the reduction in the %PPI was seen only for higher values of DA factor accompanied by the decrease in NAc activity. Thus, these results suggest that the PPI impairment caused by the dopamine agonists is dependent on the inhibition of NAc. The reduction in %PPI caused by the simulation of a dopaminergic drug in NAc was dependent on the VP activity. This was also detected in reports testing the effect of dopaminergic drugs in rats [72]. Hence, the modulation of NAc over the PPI also involves its connection with VP. Since in the present work the reduction of NAc activity was associated with an overall increase in VP activity, and the VP has an inhibitory connection with PPTg, the impact of dopaminergic transmission and NAc activity depends on the VP activation. It is interesting to note that there is a convergence of inputs from both ventral and dorsal striatum to PPTg that modulates PPI [73,74,75]. As implemented in the present model, the NAc controls the PPI tonus. Moreover, dopaminergic input to NAc encodes stimulus salience [24]. Therefore, by the NAc-PPTg connection, dopamine may regulate the sensorimotor gating tonus by increasing or decreasing PPTg activity, controlling the flow of information to other brain regions [22,25]. This pathway links higher order brain regions to regions associated with sensorial processing.

These results together show that the NAc and the VP constitute a central hub for the PPI modulation. Several brain regions are connected with the ventral striatum, as the amygdala, hippocampus, and mPFC. As shown above, the effects of amygdala manipulation are dependent on VP activity. A similar result is obtained by NAc manipulation. Since both NAc and VP are connected with PPTg, they form a bridge between cortical regions with the PPI pathway. However, studies with rats also found that the dorsal striatum and the dopaminergic transmission from substantia nigra also modulates the PPI by their connections with VP [73,76]. Disruption in striatal function could also lead to deficits in PPI by affecting the NAc- and VP-PPTg connections. Intrinsic alterations in the NAc and VP functioning or their afferences could account for the ASR and PPI alterations observed in psychiatry conditions (as schizophrenia and bipolar disorder) or neurological diseases (such as Parkinson’s disease) [41]. In particular, schizophrenia and animal models of schizophrenia have robust impairment in PPI [40] and interference in striatal function [43].

The present computational model may also be used to simulate alterations in dopaminergic transmission and striatal activity as seem in schizophrenia. Regarding these disturbances with its PPI deficits, the effect has been simulated (work in progress). Furthermore, the present model also shows that alterations in NAc, VP, and dopaminergic modulation in these regions can cause PPI deficits by the connections with the PPTg, suggesting that the PPTg is a possible target for studies evaluating strategies for recovering PPI deficits.

As mentioned above, the dorsal striatum also modulates PPI by its projections to PPTg [73,76]. Therefore, the implementation of dorsal striatum and its dopaminergic modulation is a possible extension of the present computational model. Other neuromodulators such as serotonin and adrenaline are also involved in the PPI [5,77,78]. Interestingly, the systemic administration or injection into NAc of cannabidiol attenuated the deficit caused by amphetamine in mice [79]. These neurotransmitters could be implemented by adding units representing the locus coeruleus and raphe nucleus and the connections with cortical and subcortical units, being a second extension of the present work.

### Limitations and Future Works

Some limitations should be addressed in the present work to avoid a misinterpretation of the results. First, the background sound used in the experiments was not directly implemented as did the model proposed by Schmajuk and Larrauri [51]. The model developed by these authors also showed an effect of background intensity [51] over ASR. Since in the present model we did not implement the background, it was impossible to investigate how that could interact with the tests conducted here. Accordingly, the sound intensity above the background was indirectly taken considering a background intensity fixed at 60 dB. Most of the experimental studies with GABAergic and dopaminergic drugs that we used to construct and validate this computational model used a background around 60 dB. Thus, following previous computational models [49,50], we chose to leave this factor out and adjust the model parameters to reflect the response considering a virtual background of 60 dB. Our aim was to implement a model that had the modulatory units and could account for the effects of GABAergic and dopaminergic drugs, so we focused on implementing the minimal amount of details needed to accomplish this objective. However, the model could be expanded to account for the background sound by: (1) adjusting the model parameters to decrease the units’ responsivity to the sound stimulus; and (2) implementing a similar mechanism as the one described by Schmajuk and Larrauri [51].

Second, in the model proposed here, the amygdala can increase the release of DA within NAc by a connection with VTA. Although there is a direct connection between those structures [80], some reports showed that the mechanisms for amygdala regulation of dopamine release in NAc takes place within NAc and that this regulation was independent from VTA activity [81,82]. However, there are subclasses of neurons in VTA that fire to salient or aversive events [83,84,85] and both amygdala and VTA are associated with processing stimulus salience [86,87]. Furthermore, it cannot be excluded that there is an indirect connection between amygdala and VTA. The central nucleus of amygdala, for example, was shown to send direct projections to substantia nigra in the midbrain [88]. There is also a connection between amygdala and hypothalamus [89] and from hypothalamus to VTA [90,91], suggesting a possible pathway for DA regulation by the amygdala.

In the present study, we focused on implementing the traditional units associated to the ventral striatum, namely nucleus accumbens, amygdala, and ventral pallidum. However, we could not omit a more complex scenario involving these regions with the modulation of dopamine. Both dorsal and ventral striatum show an intricate interconnection and regulation of the dopaminergic system, involving the VTA and substantia nigra, and an interaction with cortical regions [33,34]. The striatum and dopaminergic system form loops in which the ventral striatum can regulate the dopaminergic activity of both VTA and substantia nigra. From there, the dopaminergic neurons project back to the striatum [34]. Thus, this intercommunication between ventral and dorsal striatum within the dopaminergic system could also be involved in PPI modulation. Indeed, lesions of the substantia nigra pars reticulata and injection of muscimol in this region impairs PPI in rats [73,92,93]. In addition, injection of the $D_{1}$ receptor antagonist SCH23390 into the dorsal striatum reduced the PPI [76]. However, even with this modulation of PPI, the effects associated with substantia nigra alteration seem to depend on the PPTg and CPN [73,93]. Thus, the units implemented here, in particular the PPTg and CPN, are convergent units for the PPI mediation and modulation. Even if the dorsal striatum and substantia nigra were not directly implemented, in principle their effect over PPI could then rely on these two above-mentioned model units.

In addition to the dopaminergic modulation of PPI, other neuromodulators, such as noradrenaline and ATP, also influence dopaminergic transmission in the striatum. Noradrenaline controls the VTA activity and also promotes the release of dopamine from dopaminergic terminals [94]. A study associated the activation of P2Y receptors by ATP with the release of dopamine in a rat striatum [95]. Furthermore, glutamate and acetylcholine also regulate dopamine release in this region [96]. Thus, dopaminergic transmission has an intricate regulation by other neuromodulators and neurotransmitters within the striatum. These mechanisms could also be implemented in the present model, enabling the study of their interaction with the current mechanisms and their impact on PPI.

Since the present model has the flexibility to incorporate other units, such as the dorsal striatum and substantia nigra, and neuromodulators, including noradrenaline and serotonin, without changing its general form, it is possible to implement these units and investigate their roles on PPI mediation and modulation. It is important to mention, however, that the addition of new units will also make the model more complex, increasing the number of parameters that must be fitted in order to correctly replicate the experimental findings and making it more difficult to grasp an intuitive understanding and obtain insights that a simpler model can offer.

All the reports for the anatomical and functional basis of PPI considered for developing this model refer to studies with rodents. Although there is an overlap between different species in the structures involved in the generation and modulation of PPI and the effect of pharmaceutical action on the dopaminergic system [2,64,65], there are also some specific differences across species and among rodent strains in the effect of glutamatergic, dopaminergic, and serotonergic drugs [23,26,31,62,97,98]. Regarding the dopaminergic system, mice have more robust deficits in PPI for D1 receptor agonists, while rats show stronger PPI deficits with $D_{2}$ receptor agonists [31]. Similarly, although not implemented in the present model, the inhibition of substantia nigra pars reticulata decreases PPI in rats while facilitating PPI in monkeys [92]. Amphetamine administration reduces PPI in rats but causes a PPI deficit in humans only for short-intervals ISI and for within-subject analyses [99]. These differences could be associated to distinct connectivity between the dopaminergic system with cortical and subcortical regions across species [33,100], spiking pattern of dopaminergic neurons in VTA [101], and receptor binding specificities [102]. Thus, even though PPI displays similarities among different species [2], the differences between them should not be overlook in interpreting and extrapolating the results of the present model to humans and other species. Since the majority and most robust studies on the neurophysiological and anatomical basis of the ASR and PPI responses were conducted in rodents, we focused in validating the model parameters with these studies. However, it is possible, in principle, to change the model parameters to reproduce the ASR and PPI responses of other species.

Finally, the current model provides a tool for investigating how changes in simulated brain function may be linked to psychiatric disorders. We conducted different tests to understand how functional alterations could induce PPI deficits similar to those observed in patients with schizophrenia (which will be shown in a forthcoming paper). Other psychiatric disorders also display PPI deficits and our model could be extended to account for specific alterations associated with them [35,37]. For example, the implementation of dorsal striatum and substantia nigra could help in understanding their involvement in the PPI deficits observed in Parkinson’s disease [35]. Moreover, other psychiatric disorders share genetic and functional alterations with schizophrenia, such as bipolar disorder [37,103]. Thus, the present model could be used, at least in a first approximation, to study these psychiatric conditions.

## 5. Conclusions

In conclusion, the present computational model implements a neural network that has the basic features observed in animal experiments and shows a valuable approach to understanding the influence of the cortical and subcortical structures and the dopaminergic transmission over PPI. It enables the study and simulation of GABAergic and dopaminergic drugs and offers a mechanistic way to interpret their effects. According to the simulation results, NAc and VP constitutes a central hub for the PPI modulation by interfering with the PPTg activity. Ultimately, this work contributes to the understanding of how disturbances in cortical and subcortical structures can impair the sensorimotor gating.

## Figures and Tables

**Figure 1 brainsci-14-00502-f001:**
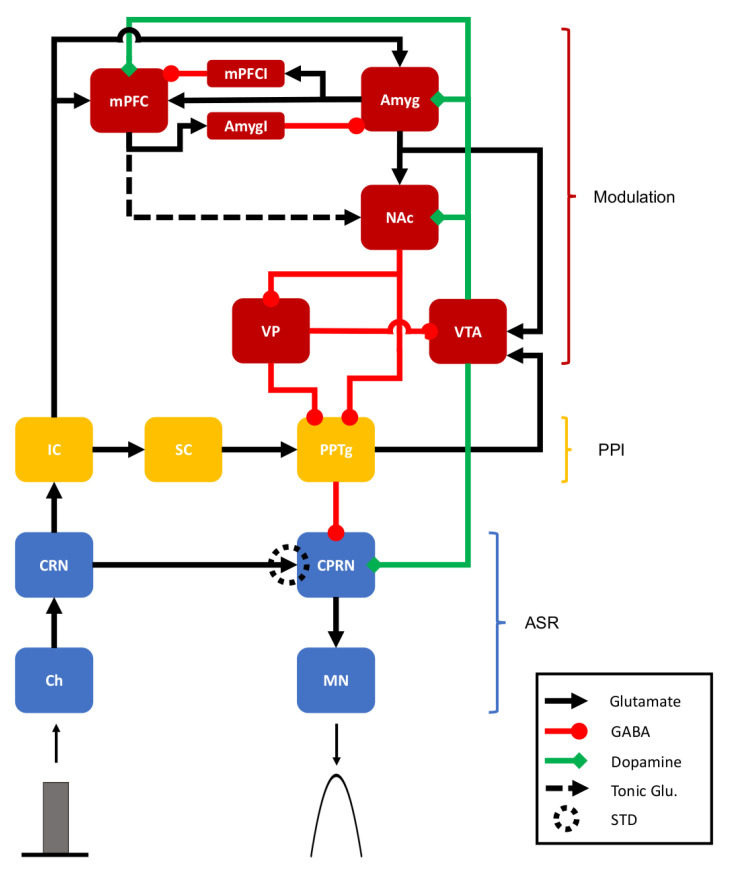
Representation of the neural network for ASR, PPI, and their modulation. The model is composed by the following structures: ASR Pathway (blue boxes): Cochlea (Ch); Cochlear Root Nucleus (CRN); Caudal Pontine Reticular Nucleus (CPRN); Motor Neuron (MN); PPI Pathway (yellow boxes): Inferior Colliculus (IC); Superior Colliculus (SC); Pedunculopontine Tegmental Nucleus (PPTg); Modulatory Pathway (red boxes): Excitatory (mPFC) and Inhibitory (mPFCI) Medial Prefrontal Cortex Subpopulations; Excitatory (Amyg) and Inhibitory (AmygI) Amygdala Subpopulations; Nucleus Accumbens (NAc); Ventral Pallidum (VP); Ventral Tegmental Area (VTA). Black lines with arrow heads represent glutamatergic excitatory projections, red lines with round heads represent GABAergic inhibitory connections, and green lines with diamond heads represent dopaminergic connections. The connection between CRN and CPRN is subjected to short-term depression (STD), as indicated by the dotted circle. The dotted black line from mPFC to NAc represents the tonic glutamatergic projection that regulates the dopaminergic system. The gray rectangle represents the sound input to the cochlea, and the parabola represents the activity of the motor neurons, which is the network output.

**Figure 2 brainsci-14-00502-f002:**
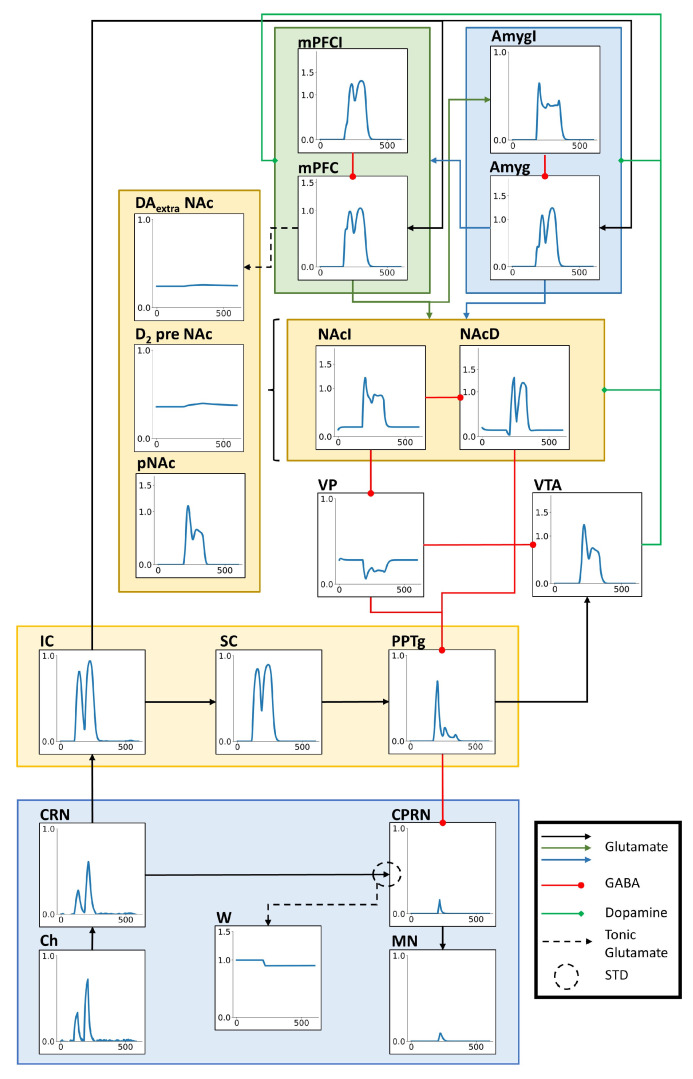
Activity of each model unit in a PPI prepulse + pulse trial. In each graph, the x axis represents the time of simulation in milliseconds (ms) and the y axis represents the unit activity (arbitrary units). The black, green, and blue arrows represent glutamatergic connections, the red arrows with circle endings represent GABAergic connections, and the green arrows with diamond endings represent the dopaminergic modulatory connections. The black dashed arrow represents the tonic glutamatergic projection from mPFC to NAc, regulating the extracellular concentration of dopamine. The dashed circle in the connection from CRN to CPRN indicates the STD mechanism between these units. In the trial simulated, the prepulse had an intensity of 25 dB above background, pulse was 60 dB above background, and ISI was 80 ms. Both stimuli had durations of 30 ms.

**Figure 3 brainsci-14-00502-f003:**
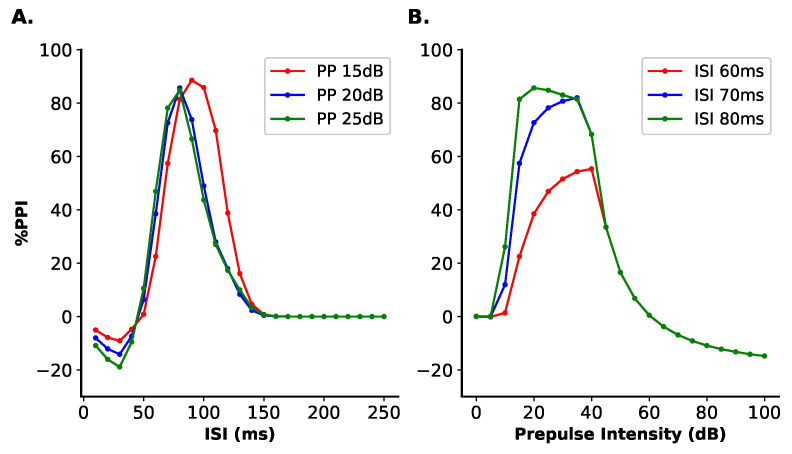
Variation of ISI and prepulse intensity on the %PPI simulation. In both tests, it was simulated trials of prepulse + pulse with pulse intensity set to 60 dB above background and pulse and prepulse duration of 30 ms. (**A**) ISI test. The ISI varied between 0 ms to 250 ms in steps of 10 ms between the simulations. This test was conducted with the prepulse intensities of 15, 20, and 25 dB above background. The maximum value of PPI is reached around 70 ms to 100 ms (depending on the prepulse intensity), below and above which the %PPI obtained in the simulation decreased, indicating an optimal ISI within this interval. (**B**) Prepulse intensity test. The ISIs used in this tests were 60, 70, and 80 ms. The prepulse intensity varied from 0 dB to 100 dB above background in steps of 5 dB between the simulations. The %PPI increases as the prepulse intensity increases from 0 dB and reaches a maximum %PPI for intensities between 20 dB and 40 dB (depending on the ISI). Above these intensities, the %PPI starts to decrease and above 60 dB there are negative values of %PPI.

**Figure 4 brainsci-14-00502-f004:**
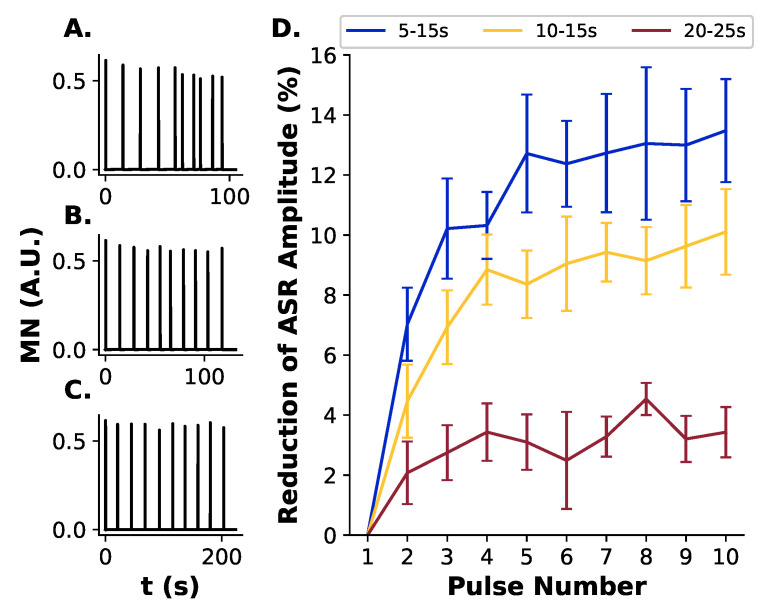
Habituation of ASR units. Repeated pulse presentation led to a reduction in ASR amplitude (MN activity). The test session was composed of 10 pulse trials and ITIs of 5–15 s, 10–15 s, and 20–25 s. For each ITI, ten simulations were ran with randomly selected model parameters. The percentage of decrease in MN activity was calculated related to the first pulse presentation. (**A**) Example of a simulation with ITI of 5–15 s. (**B**) Example of a simulation with ITI of 10–15 s. (**C**) Example of a simulation with ITI of 20–25 s. (**D**) Percentage of reduction in ASR Amplitude (MN activity) by repeated pulse presentation for each ITI tested. The MN activity reductions by the repeated pulse presentation were different across the ITIs used. There was a higher percentage reduction in MN activity for the ITI of 5–15 s (maximum reduction of 11.14%) compared to the ITI of 10–15 s (maximum of 8.31%) and 20–25 s (3.69%).

**Figure 5 brainsci-14-00502-f005:**
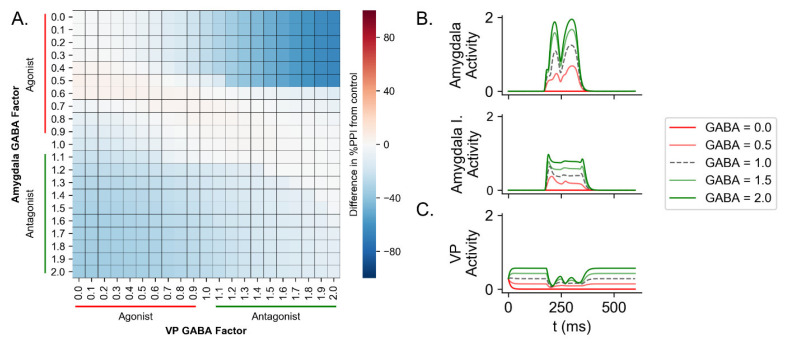
Effect of the manipulation of GABA factor in amygdala and VP over the %PPI and their activity in a trial of PPI. For the GABA factor between 0.00 and 1.00, there was an inhibition of the unit under test (simulating the effect of a GABAergic agonist) and for values between 1.00 and 2.00, there was a hyperactivation of the unit (simulating the effect of a GABAergic antagonist). This test was conducted by varying the GABA factor in the amygdala (Equations (Equation 18) and (Equation 17) in the Appendix A) and in VP (Equation (Equation 28) in the Appendix A)) from 0.00 to 2.00 in steps of 0.10 between simulations. We simulated trials of prepulse + pulse with pulse intensity of 60 dB above background, prepulse intensity of 25 dB, ISI of 80 ms and duration of prepulse and pulse of 30 ms. (**A**) The horizontal axis indicates the value of the GABA factor in the VP. The vertical axis indicates the value of the GABA factor in the amygdala. The horizontal line at 1.00 for the GABA factor in the amygdala corresponds to the manipulation of only VP activity. The vertical line at 1.00 for the GABA factor in VP corresponds to the manipulation of only amygdala activity. The results are shown as difference in %PPI from control (GABA factor at 1.00 in both amygdala and VP). Negative values for %PPI indicate less inhibition compared to control, while positive values indicate more inhibition. The inhibition and hyperactivation of amygdala and the inhibition of VP cause a reduction in the %PPI obtained in the simulation. The combination of amygdala inhibition and VP hyperactivation, amygdala hyperactivation, and VP inhibition or amygdala hyperactivation and VP hyperactivation caused a further decrease in the %PPI compared to the manipulations of amygdala alone or VP alone. Interestingly, the inactivation of both amygdala and VP attenuated the reduction in %PPI caused by the inhibition of amygdala alone or VP alone. (**B**) Activities in the excitatory (Amygdala) and inhibitory (Amygdala I.) subpopulations in amygdala for five different values of GABA factor in amygdala. (**C**) VP activity for five different values of GABA factor in VP.

**Figure 6 brainsci-14-00502-f006:**
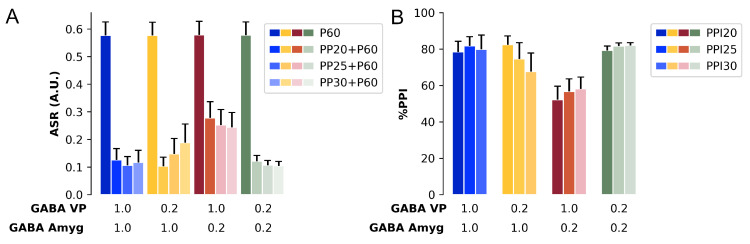
Simulation of PPI sessions under amygdala and/or VP inhibition. The test session was composed by 74 stimuli presentations including pulse (60 dB above background), prepulse (15, 20 and 25 dB above background), prepulse + pulse, and background. The ITI was set to 5–15 s, the ISI of 80 ms, and duration of pulse and prepulse of 30 ms. Columns represent the groups: VP group (GABA factor at 0.20 in VP), the Amygdala group (GABA factor at 0.20 in amygdala), Amygdala-VP group (GABA factor at 0.20 in both VP and amygdala), and Control group (GABA factor at 1.00 in both units). (**A**) ASR as mean±SEM. The ASR amplitude for the pulse was higher compared to the ASR amplitude for all the prepulse intensities. The Amygdala group showed higher ASR for PP15 + P60 and PP20 + P60 than all other groups and ASR for PP30 + P60 higher than the Control and Amygdala-VP group. (**B**) %PPI as mean±SEM. The %PPI of the Amygdala group was lower than the control condition. There was no difference between the Amygdala-VP group compared to the control condition.

**Figure 7 brainsci-14-00502-f007:**
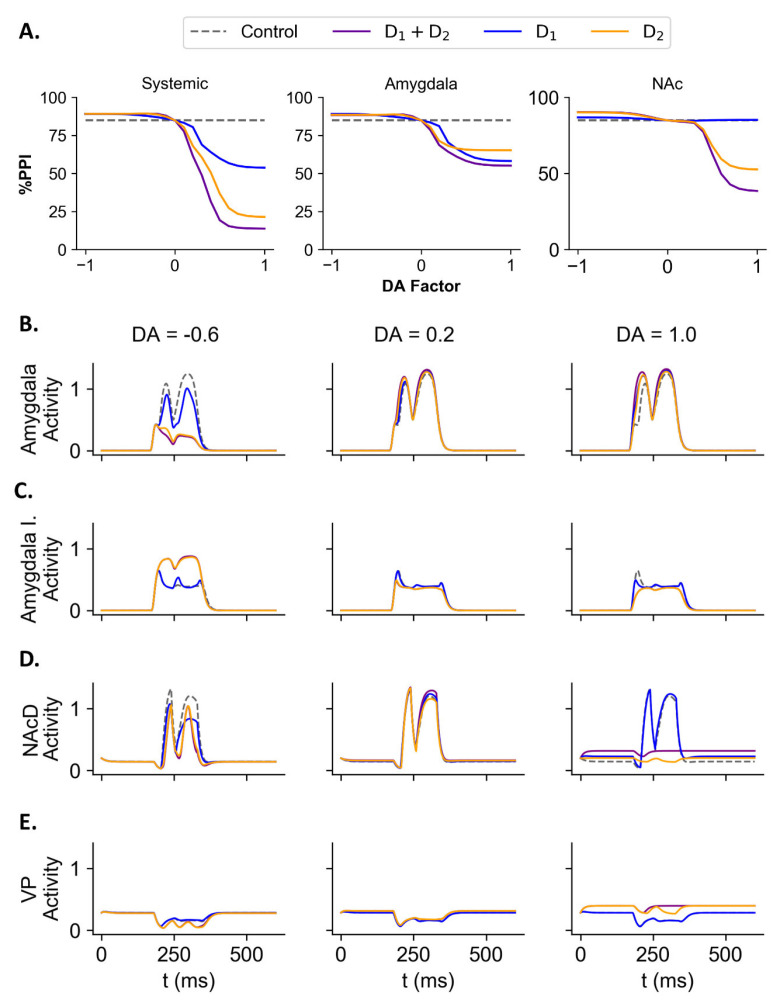
Simulation of DA factor manipulation on a PPI trial. The trials simulated had a pulse intensity of 60 dB, prepulse intensity of 25 dB, ISI of 80 ms and duration of stimuli of 30 ms. When −1.00≤DA<0.00, it was simulated the effect of a dopaminergic antagonist, while for 0.00<DA≤1.00 it was simulated the effect of an agonist. The control condition is defined as the DA factor at 0.00. The purple line indicates the tests with manipulation of both dopaminergic receptor types. The blue line represents the tests with manipulation of only $D_{1}$ receptors. The orange line indicates the tests with manipulation of DA factor only in $D_{2}$ receptors. (**A**) Effect of manipulation of DA factor in all the dopaminergic receptors in the model (systemic test), in amygdala, and in NAc. With exception for the D_1_ receptor in NAc, there was a reduction in the %PPI with the increase of DA factor compared to control. The increase of DA factor in D_1_ receptors in NAc caused a slight increase in the %PPI compared to control. In all cases, decreasing the DA factor led to an increase in the %PPI compared to control. (**B**) Amygdala activity for three values of DA factor (−0.6, 0.2 and 1.0) in $D_{1}$, $D_{2}$ or both receptors in amygdala. Dashed line represents the control condition (DA factor = 0.0). The decrease in DA factor led to a reduction in amygdala activity, while the increase in DA factor caused an increase in the activity. (**C**) Activity of inhibitory subpopulation in amygdala for three values of DA factor (−0.6, 0.2 and 1.0) in $D_{1}$, $D_{2}$ or both receptors in amygdala. Dashed line represents the control condition (DA factor = 0.0). The decrease in DA factor in amygdala led to an activity increase in the inhibitory subpopulation in amygdala. The increase in DA factor caused a slight decrease in the activity of the amygdala inhibitory subpopulation. (**D**) NAc activity for three values of DA factor (−0.6, 0.2, and 1.0) in $D_{1}$, $D_{2}$, or both receptors in NAc. Dashed line represents the control condition (DA factor = 0.0). There was a decrease in NAc activity by decreasing the DA factor in $D_{1}$, $D_{2}$, or both receptors. For a DA factor at 0.2, there was a slight increase in NAc activity. In contrast, for a high DA factor (at 1.0), there was a reduction in NAc activity. (**E**) VP activity for three values of DA factor (−0.6, 0.2 and 1.0) in $D_{1}$, $D_{2}$, or both receptors in NAc. Dashed line represents the control condition (DA factor = 0.0). For the DA factor at 1.0 of $D_{2}$ or both receptors in NAc there was an increase in VP activity compared to control.

**Figure 8 brainsci-14-00502-f008:**
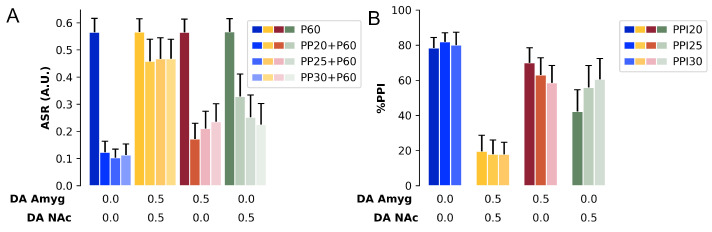
Simulation of the increased activity of $D_{1}$ and $D_{2}$ receptors. The test session was composed of 74 trials among pulse (60 dB above background), prepulse (15, 20, and 25 dB above background), prepulse + pulse, and background. The ITI was set to 5–15 s, the ISI of 80 ms, and duration of pulse and prepulse of 30 ms. The groups were as follows: Systemic group (DA factor at 0.50 in both $D_{1}$ and $D_{2}$ receptors and in all receptors of every unit that received the dopaminergic input), Amygdala group (DA factor at 0.50 in both $D_{1}$ and $D_{2}$ receptors in amygdala), NAc group (DA factor at 0.50 in both $D_{1}$ and $D_{2}$ receptors in NAc), and Control group (DA factor at 0.00 in all receptors). (**A**) ASR as mean±SEM. The ASR amplitude for the pulse was higher than all ASR amplitude for any prepulse intensities. The systemic group also showed a higher ASR for all prepulse + pulse trials compared to Control, Amygdala, and NAc groups. (**B**) %PPI as mean±SEM. The %PPIs in Systemic and NAc were lower than the control condition for all prepulse intensities. Except for the PP15, the Amygdala showed lower %PPI than the Control condition. The amygdala and NAc groups showed higher %PPI than the Systemic group for all prepulse intensities.

## Data Availability

The codes used to implement and simulate the model can be found in: https://github.com/ThiagoTakechi/PPIModel.git, accessed on 13 May 2024.

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
