# Peer review of "A Computational Model for the Simulation of Prepulse Inhibition and Its Modulation by Cortical and Subcortical Units"

_brainsci, 2024, doi:10.3390/brainsci14050502_

Round 1

Reviewer 1 Report

Comments and Suggestions for Authors

A Computational Model for the Simulation of Prepulse Inhibition and Its Modulation by Cortical and Subcortical Units

 This is an interesting simulation study. The authors can find my appraisal, recommendations, and commentaries section by section as follows:

Introduction: This section needs to be organized in a better way. I suggest to introduce the PPI concept, then the neurophysiology, and then pathology (mainly psychiatric disorders as reported by the authors).   The authors introduced the concept of PPI. However and correctly, they stated that PPI showed alterations in psychiatric disorders, with only one reference about PPI in psychiatric disorders apart from schizophrenia, then a wide explanation was given. This concept needs to be discussed in a better way. For example, adding more info about specific psychiatric disorders and the specific alterations of PPI. Similarly, I advise explaining in a better way the link between neurotransmitter alterations, apart from dopamine,  and psychiatric disorders.

For example, NA is an interface between dopaminergic, and opioidergic system.  A discussion about it should be interesting.

 Moreover, the role of dopamine is not clear and briefly mentioned in the text.

I suggest to add a brief discussion about the dopamine receptors.

A meta-analysis is also a systematic review, but I suggest using the term meta-analysis.

The methods are well written and sufficiently explicative of the procedure. The values of the ISI need to be justified. The authors stated, “We calculated the %PPI for each ISI and prepulse intensity in these tests.” Line 120. This statement needs to be justified. It is not clear if the authors calculated tests fo heteroscedasticity, normality etc. before to apply the ANOVAs (please, add in the text).

The results are very complex and a bit difficult to follow, but according to me, despite the complexity, are very interesting. I also agree with the structure of the section and it is intelligible.  However, figure 2 needs to be improved, and I suggest naming each graph and adding the title.

According to me, the discussion integrates previous findings with study ones. 

Reviewer 2 Report

Comments and Suggestions for Authors

The article introduces an innovative computational model that replicates prepulse inhibition (PPI) and investigates the impact of several parameters on this phenomenon. The model exhibits various strengths, such as its capability to accurately reproduce essential experimental characteristics of protein-protein interactions (PPI) and its ability to simulate the impacts of pharmaceutical interventions. Some of the major concerns that need to be addressed are given below.

·         The current model does not include background sound, which is a significant element in real-world (ASR). Integrating ambient noise would enhance the authenticity of the model and enable examination of its interaction with PPI. Is there any possibility of including the same in your future?

·         The present oversimplification of the amygdala-NAc-dopamine relationship may not fully encompass the reported complexities. To enhance the model's biological plausibility, it is important to consider and account for the challenges involved, such as possible indirect pathways that may involve the substantia nigra.

·         The model is derived from experiments conducted on rodents. The authors should highlight the potential constraints in extrapolating the findings to other species, including humans. It would be advantageous to briefly examine the known variations in PPI and dopaminergic function among different animals.

·         The discussion might be extended to provide further details on future avenues of research utilising the model. This may involve replicating changes in schizophrenia to understand PPI impairments, integrating additional neuromodulators, and investigating the model's suitability for various animals.

Round 2

Reviewer 1 Report

Comments and Suggestions for Authors

The Authors addressed all my concerns.